# Protective Effect of Probiotic Bacteria and Estrogen in Preventing HIV-1-Mediated Impairment of Epithelial Barrier Integrity in Female Genital Tract

**DOI:** 10.3390/cells8101120

**Published:** 2019-09-21

**Authors:** Sara Dizzell, Aisha Nazli, Gregor Reid, Charu Kaushic

**Affiliations:** 1Department of Pathology and Molecular Medicine, McMaster University, Michael G. DeGroote Center for Learning and Discovery, Hamilton, ON L8P 3Z5, Canada; sara.dizzell@gmail.com (S.D.); nazlia@mcmaster.ca (A.N.); 2McMaster Immunology Research Center, McMaster University, Michael G. DeGroote Center for Learning and Discovery, Hamilton, ON L8S 4L8, Canada; 3Departments of Microbiology & Immunology and Surgery, Western University, and Canadian Research and Development Centre for Human Microbiome and Probiotics, The Lawson Health Research Institute, London, ON N6A 4V2, Canada; gregor@uwo.ca

**Keywords:** HIV-1, probiotic, estrogen, progesterone, genital epithelial cells, barrier

## Abstract

Approximately 40% of global HIV-1 transmission occurs in the female genital tract (FGT) through heterosexual transmission. Epithelial cells lining the FGT provide the first barrier to HIV-1 entry. Previous studies have suggested that certain hormonal contraceptives or a dysbiosis of the vaginal microbiota can enhance HIV-1 acquisition in the FGT. We examined the effects of lactobacilli and female sex hormones on the barrier functions and innate immune responses of primary endometrial genital epithelial cells (GECs). Two probiotic strains, *Lactobacillus reuteri* RC-14 and *L. rhamnosus* GR-1, were tested, as were sex hormones estrogen (E2), progesterone (P4), and the hormonal contraceptive medroxyprogesterone acetate (MPA). Our results demonstrate that probiotic lactobacilli enhance barrier function without affecting cytokines. Treatment of GECs with MPA resulted in reduced barrier function. In contrast, E2 treatment enhanced barrier function and reduced production of proinflammatory cytokines. Comparison of hormones plus lactobacilli as a pre-treatment prior to HIV exposure revealed a dominant effect of lactobacilli in preventing loss of barrier function by GECs. In summary, the combination of E2 and lactobacilli had the best protective effect against HIV-1 seen by enhancement of barrier function and reduction in proinflammatory cytokines. These studies provide insights into how probiotic lactobacilli in the female genital microenvironment can alter HIV-1-mediated barrier disruption and how the combination of E2 and lactobacilli may decrease susceptibility to primary HIV infection.

## 1. Introduction

Currently, women account for more than 50% of people infected with HIV-1 (human immunodeficiency virus-1) worldwide [1]. Globally, the number of new HIV-1 infections reported in women has risen continuously over the past few decades and women have been shown to be at an increased risk of HIV-1 infection compared to men [1]. Infection in women occurs primarily through heterosexual transmission in the female genital tract (FGT) with early mechanisms still not well understood [2]. 

Although the etiology of greater acquisition within the FGT is not fully understood, increased inflammation in the FGT has been suggested as a common theme associated with higher rates of transmission [3,4,5,6,7]. The FGT can be divided into two distinct compartments. The lower FGT consists of the vagina and the ectocervix and is protected by multiple layers of stratified squamous epithelium. In contrast, the upper FGT consists of the endocervix, uterus, fallopian tubes and ovaries, and is lined by a single layer of columnar epithelium connected by tight junction proteins. Although the tight junctions help to strengthen the epithelium, the upper FGT is still vulnerable to HIV infection, as it is composed of a single layer of cells, thus placing the virus in closer proximity to target cells beneath the epithelial layer. Studies have demonstrated that upper genital tract, especially the transition zone between ecto and endo cervix is highly susceptible to HIV-1 infection [3,8,9]. In order for HIV-1 to establish successful infection, the virus must cross the epithelial barrier in order to access target cells [10]. Previous studies have demonstrated that the HIV-1 surface glycoprotein-120 (gp120) directly interacts with primary human genital epithelial cells (GECs), leading to the production of proinflammatory cytokines, including tumor necrosis factor-alpha (TNF-α). This production decreases GEC barrier integrity and subsequent microbial translocation of HIV-1 across the upper female genital tract epithelium [11,12]. 

A significant body of research has suggested that female sex hormones, progesterone (P4) and estrogen (E2), play a role in HIV-1 infection in women, as they have been shown to regulate innate immunity and inflammation in the FGT [4,13,14,15,16]. Furthermore, over 100 million women globally use hormonal contraceptives (HCs) [17] and a growing body of literature suggests there may be an increased risk of HIV-1 acquisition caused by use of injectable HCs [18,19,20]. Of particular interest is the progestin-based HC, Depot medroxy-progesterone acetate (DMPA or Depo-Provera). The low cost, long-term effectiveness of DMPA as a contraceptive makes it an attractive choice for women. Notably, DMPA is most often used in the developing world, especially in regions such as sub-Saharan Africa, where HIV-1 prevalence is at its highest. Human epidemiologic studies have shown an association between DMPA use and increased HIV-1 infection, disease progression and mortality [21,22]. Additionally, several meta-analyses have examined the strength of the evidence in light of conflicting evidence regarding the use of HCs and HIV-1 acquisition [19,20,21,23]. One such analysis found that DMPA increased women’s risk of contracting HIV-1 by 40 percent compared to those not using the contraceptive [19]. This was confirmed in two further meta-analyses showing a 49 and 40 percent increased risk of HIV-1 acquisition with DMPA use [23,24,25]. Although associations have been observed between the use of injectable HCs and increased HIV-1 susceptibility, the pathways involved in these outcomes remain unclear and are under-investigated.

In addition to hormones, the microbiota of the FGT has been correlated with alterations in HIV-1 susceptibility. The composition of the microbiota within the FGT influences HIV-1 susceptibility and may be regulated by female sex hormones [13,25,26]. Molecular studies have identified more than 250 microbial species capable of growth within the vaginal tract [27]. A decreased abundance of microbial species has also been identified in upper genital tract [28]. Factors such as age, ethnicity, female sex hormones and use of HC have been shown to affect the composition and relative abundance of these species [29]. In the majority of women, a ‘healthy’ FGT microbiota is characterized by the abundance of *Lactobacillus* species, mainly, *L. crispatus, L. jensenii, L. gasseri* and/or *L. iners* [28,30]. However, when the composition of the microbiota is shifted toward polymicrobial species and increased bacterial diversity, it is associated with bacterial vaginosis (BV) [27]. A decrease in *Lactobacillus* dominance and increased diversity is commonly associated with increased susceptibility to STIs including HIV-1, HSV-2, gonorrhea, and Chlamydia [31,32]. In comparison, a *Lactobacillus*-rich environment exerts protective effects against urogenital diseases, including, BV, yeast infections, HSV-2, and HIV-1 [33]. Interestingly, it has been suggested that the menstrual cycle may be associated with changes in the FGT microbiota [13]. Cyclic changes observed with the differing phases of the menstrual cycle may involve factors such as hormones, pH and glycogen content, bacterial adherence, their ability to colonize epithelial cells, and the composition of microbial species [34,35]. 

The concept of applying probiotic lactobacilli to the vagina to displace pathogens and improve host defenses dates back to the mid-1980s but has more recent interest [34]. The most documented probiotic strains for urogenital application, *L. rhamnosus* (GR-1) and *L. reuteri* (RC-14), were selected for testing in these studies due to these and other attributes, and their benefits were shown in a series of clinical studies. These include showing that the acid produced by both strains could kill HIV [35], and the instillation of GR-1 into the vagina up-regulated host defenses and barrier function [36]. 

Since the upper genital tract epithelial barrier is most susceptible to HIV-1, the aim of this study was to examine the interactions between these epithelial cells, sex hormones and probiotic strains of bacterial in the context of HIV-1 infection. The results reveal that a combination of lactobacilli and estrogen provides both enhanced barrier function and reduced proinflammatory cytokine production by GECs in response to HIV-1.

## 2. Materials and Methods

### 2.1. Isolation and Culture of Primary Genital Tract Epithelial Cells

Primary genital epithelial cells were isolated from cervical and endometrial tissues obtained from women aged 30–59 years (mean age 42.9 ± 7.2) undergoing hysterectomies for benign gynecological reasons at Hamilton Health Sciences Hospital. Informed written consent was obtained in accordance with the approval of the Hamilton Integrated Human Research Ethics Board. A detailed protocol for the isolation and culture of primary GECs has previously been described by Kaushic et al., [26]. Briefly, endometrial or endocervical tissues were minced into small pieces, digested in an enzyme mixture for an hour at 37 °C. GECs were isolated by a series of separations through nylon mesh filters (Small Parts, Miramar, FL, USA), resuspended in DMEM/F12 primary growth medium (Invitrogen, Burlington, ON, Canada), and seeded onto Matrigel-coated (BD Biosciences, Mississauga, ON, Canada) tissue culture inserts (BD Biosciences). GEC cultures were grown for 5–7 days until confluent monolayers were formed. The confluency was monitored by transepithelial resistance (TER) measured by a volt ohm meter (EVOM; World Precision Instruments, Sarasota, FL, USA). Confluent monolayers showing TER values greater than 1000 Ω/cm^2^ were used for further experiments. The purity of epithelial monolayers was between 95% and 98%, with no trace of any hematopoietic cells.

### 2.2. Hormone and Contraceptive Treatment

Sex hormones E2 (10^−3^ M), P4 (10^−4^ M) (purchased as water-soluble products, Sigma Aldrich, Oakville, ON, Canada), or MPA (10^−3^ M) (Sigma Aldrich) were prepared as stock in either water (E2, P4) or 100% ethanol (MPA) and diluted to the final working concentrations 10^−9^ M, 10^−7^ M and 10^−9^ M, respectively, in phenol-red–free Dulbecco’s modified Eagle’s medium/F12 (Life Technologies, Burlington, ON, Canada), the primary cell media [37]. The final dilution of ethanol in the primary cell medium was 10^−6^ fold. Sex hormone concentrations correspond to highest serum levels during menstrual cycle.

### 2.3. Treatment of GECs with Lactobacilli and HIV-1 Exposure

The probiotic *Lactobacillus reuteri* RC-14 and *Lactobacillus rhamnosus* GR-1 were provided by Dr Gregor Reid, Lawson Health Research Institute, London, Ontario [35]. Both strains were grown anaerobically in MRS broth (VWR, Mississauga, ON, Canada) at 37 °C. Prior to the addition of bacteria to GECs, the RC-14 and GR-1 were washed to remove the MRS medium. The bacteria were then re-suspended in primary cell culture medium and added to the GEC monolayers at a concentration of 100 CFU/cell. Following 2 h of incubation, unattached bacteria was removed, and the GEC monolayers were washed with PBS and fresh medium was added. 

HIV-1 virus strain ADA (M-tropic) was propagated by infection of adherent monocyte derived macrophages isolated from human PBMCs, followed by concentration of the virus using the Amicon Ultra-15 filtration system (Millipore, Billerica, MA, USA). Viral stocks were titered for infectious viral units per milliliter using the TZM-bl indicator cell assay as previously described [38]. HIV-1 was added as 10^5^ infectious units in 100 µL quantity on apical side of confluent GEC cultures. GECs were incubated with virus for 24 h.

### 2.4. Fluorescein Isothiocyanate (FITC)-Dextran Dye Assay for Cell Monolayer Permeability Assay 

FITC-dextran (4 kDa; 2.3 mg/mL; Sigma-Aldrich) was added to the apical surface of GECs immediately after bacteria, or during HIV incubations, and cells were incubated for 24 h, after which 50 µL of basolateral medium was sampled and placed in duplicate in a 96 well plate. FITC fluorescence was measured using a microplate reader (SpectraMax i3, Molecular Devices, Sunnyvale, CA, USA) at an excitation of 490 nm and emission of 520 nm. The FITC-dextran in the basolateral compartment is expressed as a percentage of FITC-dextran added to the apical compartment. 

### 2.5. Cell Stress and Cytotoxicity Measured by Lactate Dehydrogenase Release

Lactate dehydrogenase (LDH) enzyme was detected in the supernatants using an LDH kit (Pierce, Thermo Scientific, San Jose, CA, USA) according to manufacturer instructions. 

### 2.6. Magpix Multi-Analyte Assay for Measurement of Cytokines and Chemokines

Supernatants from GEC cultures were collected at various time points following exposure to hormones, *Lactobacillus* strains GR-1 or RC-14, or HIV-1. The Magpix multi-analyte magnetic bead-based assay was used to measure the concentrations of cytokines and chemokines including TNF-α, IL-1α, IL-8 and RANTES in supernatants, using the Magpix technology system (Millipore, Billerica, MA, USA), as per the manufacturer’s instructions. 

### 2.7. Immunofluorescent Staining of Epithelial Cell Cultures

After treatment with lactobacilli, hormone and/or HIV-1, GECs were fixed in 4% paraformaldehyde (PFA; Electron Microscopy Sciences, Hatfield, PA, USA), permeabilized, and stained for ZO-1, and immunofluorescence microscopy was performed, as previously described [11]. Images were acquired using an inverted laser-scanning microscope (Olympus, FLUOVIEW FV3000) equipped with argon laser (450 to 514 nm). Immunoreactive ZO-1 were excited using the 488-nm laser and collected using a standard fluorescein isothiocyanate filter set. An x63 oil immersion objective lense with 2X zoom was used. For each experiment, image acquisition (i.e., confocal microscope settings) and processing was identical between controls and treated cells. Images are presented en face to illustrate the distribution of the tight junction protein stained by Alexa Fluor 488 secondary antibodies (green). Fluorescence measurement on three different images of same treatment were analyzed by image analysis software (Image J version 1.50i, NIH) to measure the areas of fluorescently stained ZO-1. 

### 2.8. Statistical Analysis

Statistical analysis and graphical representation were performed using Prism 6.0d software (GraphPad Software). A one-way analysis of variance (ANOVA) using the Bonferonni post-test was also used for analysis of data from all experiments. When comparing two treatment groups, an unpaired *t*-test was used for analysis.

## 3. Results

### 3.1. Exposure to Probiotic Strains of Lactobacilli Improves the Barrier Function of Primary Upper Genital Tract Epithelial Cells

The relationship between upper genital tract epithelial cells and lactobacilli with respect to barrier function has not been well characterized. To study the interaction of the probiotic lactobacilli strains *L. rhamnosus* (GR-1) and *L. reuteri* (RC-14) with GECs, we exposed confluent monolayers of primary human endometrial epithelial cells to each bacterial strain and assessed barrier function by measuring transepithelial electrical resistance (TER) and leakage of FITC-dextran. The two strains of lactobacilli were added for 2 h, non-adherent bacteria were removed, and fresh medium was added. After a further 24 h of culture, barrier function was assessed by FITC-dextran leakage and TER, a measure of cell monolayer integrity that is commonly used as a proxy indicator to monitor the growth and condition of polarized monolayers [36]. The presence of either probiotic lactobacilli strains significantly increased TER of GEC monolayers in comparison to the no bacteria control, indicative of an enhancement in the integrity of the GEC monolayer (Figure 1A). Epithelial cell monolayer permeability to FITC-dextran is an additional measure of GEC barrier function [36]. Since the TER of GECs increased in the presence of probiotic lactobacilli, permeability was then measured to assess the ‘leakiness’ of the GEC monolayers. The 4 kDa, fluorescent FITC-dextran dye was added to the apical compartment of the GEC monolayer and the amount of dye that leaked to the basolateral compartment was measured 24 h later. There was a significant decrease in FITC-dextran that leaked to the basolateral compartment beneath the GEC monolayers treated with probiotic lactobacilli, compared to the control cultures without bacteria (Figure 1B). Both decrease in permeability and the increase in TER indicates that probiotic lactobacilli were able to enhance GEC barrier functions.

Lactate dehydrogenase (LDH) is released into the supernatant when cell membrane integrity is compromised and measured in culture supernatants as an indicator of cellular cytotoxicity and cytolysis [39]. LDH was only present in a small amount in GEC monolayers treated with the probiotic lactobacilli compared to lysed GEC cultures. No significant differences in LDH production were observed between the control treatment (no bacteria added) and cells exposed to probiotic bacteria strains (Figure 1C). This suggests that the presence of probiotic bacterial strains, GR-1 and RC-14, do not adversely affect GEC viability. 

Exposure to bacteria such as Lactobacilli may result in the production of cytokines and chemokines by GECs, thereby altering the local tissue immune environment that would impact response to pathogens. To assess the effect of lactobacilli on innate mediators of inflammation produced by GECs, we measured the production of the proinflammatory cytokines (TNF-α, and IL-1α) and the chemokines IL-8 and RANTES following lactobacilli treatment. Treatment with GR-1 and RC-14 did not significantly alter production of TNF-α, IL-1α, IL-8 or RANTES when compared to untreated GECs indicating that these strains of lactobacilli do not influence cytokines or chemokines produced by cultured GECs (Figure 1D–G). Baseline cytokine levels were found to vary between different tissue samples, but in all tissues, cytokine trends were found to be the same after treatments.

### 3.2. Estrogen Enhances Barrier Function and Modulates Proinflammatory Cytokine and Chemokine Response in Primary GECs

Many studies have demonstrated that female sex hormones regulate immune responses in the FGT [40], however the response of GECs to these hormones remain unclear. Epithelial cells in the FGT are under constant regulation by fluctuations of the sex hormones, E2 and P4, during the menstrual cycle. Additionally, hormonal contraceptives, such as MPA, influence the menstrual cycle and alter the natural hormone regulation, which may affect the function of the GECs. To better understand the role of sex hormones in the FGT, we examined how E2, P4, and MPA influence the FGT epithelial barrier functions and modulate epithelial proinflammatory responses. GECs grown in the presence sex hormone E2 showed significant increase in TER while cells grown in presence of MPA displayed significantly lower TER values compared to other treatments over time (Figure 2A). GECs treated with MPA or P4 also displayed an increase in permeability to FITC-dextran, compared to E2 and no hormone control GECs (Figure 2B). Leakage in the MPA treated monolayers was the highest of all treatments.

To ensure that the increase in permeability and decrease TER values of GECs treated with MPA was not due to decreased viability, LDH released by GECs was measured in supernatants from all treatment groups (Figure 2C). In comparison to the control of directly lysed non-treated GECs, the E2, P4, MPA and no hormone GECs demonstrated significantly less release of LDH. No significant differences were observed between the MPA treated cells and E2, P4, and no hormone. GECs grown in the presence of E2 showed a significant decrease in TNF-α, IL-1α, and IL-8 production compared to no hormone treated cells, indicating that E2 was conferring an anti-inflammatory environment (Figure 2D–F). While cells treated with P4 or MPA showed significant reduction in IL-1α production, there was no significant reduction in TNF-α or IL-8 production by those cells. None of the treatment groups had significant changes in RANTES production, although the MPA-treated group showed the highest levels of RANTES production.

### 3.3. Pre-Exposure of GECs to Probiotic Lactobacilli Prevents HIV-1-Mediated Barrier Disruption and Downregulates Proinflammatory Cytokine Production by GECs

A *Lactobacillus* rich environment has been shown to exert protective effects against urogenital diseases, including BV, yeast infection, HSV-2, and HIV-1 [33]. To further elucidate the mechanisms by which lactobacilli confer protection against HIV-1 infection, we exposed GECs to HIV-1 following pre-treatment with probiotic strains of lactobacilli. HIV-1 exposure alone resulted in reduced TER. However, this decrease was reversed in GEC cultures exposed to HIV-1 after pre-treatment with either of the two strains of lactobacilli (Figure 3A,B). Likewise, GECs treated with probiotic lactobacilli followed by HIV-1 had significantly decreased FITC-dextran dye leakage when compared to GECs treated with HIV-1 alone (Figure 3C,D). Together this data suggests that prior exposure to probiotic lactobacilli can prevent GEC barrier dysfunction caused by HIV-1 [11].

We then examined the effect of pre-exposure to lactobacilli on the proinflammatory cytokines TNF-α, IL-1α, IL-8 and RANTES produced by HIV-1 exposed GECs. Pre-treatment with GR-1 and RC-14 significantly reduced HIV-1-mediated induction of TNF-α (Figure 3E). A similar effect was observed for IL-1α, although the increase in production of this cytokine in response to HIV-1 did not reach statistical significance (Figure 3F). In contrast, production of IL-8 by GECs was unaffected by HIV-1 exposure and remained unaltered following pre-exposure to probiotic lactobacilli (Figure 3G). RANTES production was significantly increased by GECs treated with HIV-1 alone. However, pre-treatment with either probiotic lactobacilli resulted in significant reduction in RANTES production, although at levels still above that of the untreated control (Figure 3H). Thus, prior exposure to probiotic lactobacilli appeared to reduce production of some proinflammatory cytokines by GECs in response to HIV-1.

### 3.4. Estrogen Pre-Treatment Prevents Leakage and Decreases TNF-α Production but not RANTES Production Induced by HIV-1

Several studies using animal models or ex vivo cervical explants have shown that E2 plays a protective role in HIV-1 infection, while P4 increases susceptibility to HIV-1 [41,42]. Furthermore, an increased susceptibility to sexually transmitted infections (STIs) including HIV-1 has been reported in women using the hormonal contraceptive DMPA [8,21,22,43]. The effect of E2 and MPA on the GEC barrier, and modulation of immune responses as a mechanism of altered susceptibility to HIV-1, requires further investigation. Therefore, we grew GECs in the presence or absence of E2, P4 or MPA and then exposed the cells to HIV-1. Exposure of GECs to HIV-1 significantly decreased TER irrespective of prior hormonal treatment (Figure 4A). GECs grown in the absence of hormone, or the presence of P4 or MPA, showed a significant increase in FITC-dextran leakage after exposure to HIV-1 (Figure 4B). However, no significant increase in FITC-dextran leakage was observed in GECs grown in the presence of E2 and then exposed to HIV-1. Prior treatment with MPA appeared to increase the effects of HIV-1 on FITC-dextran leakage (Figure 4B).

GECs grown in the absence of hormones or pre-treated with P4 showed a significant increase in TNF-α after exposure to HIV-1, whereas pre-treatment with E2 and MPA resulted in reduced response to HIV-1 (Figure 4C). A statistically significant reduction in production of IL-1α was observed for GECs treated with E2 and MPA compared to no hormone in response to HIV-1 (Figure 4D). No significant increase in IL-8 production by GECs following HIV-1 exposure was observed in the absence or presence of hormones (Figure 4E). However, prior treatment with MPA resulted in a significant increase in IL-8 production after HIV-1 exposure. Finally, production of RANTES was examined. Regardless of the presence or absence of E2, P4 or MPA, exposure of GECs to HIV-1 resulted in significant increase in the production of RANTES (Figure 4F).

### 3.5. Probiotic Lactobacilli and Estrogen Prevent HIV-1-Mediated Epithelial Barrier Disruption

The results described thus far indicate that both E2 and probiotic lactobacilli reduce the adverse effects of HIV-1 on GEC barrier function and production of proinflammatory cytokines. Next we examined how the combination of probiotic lactobacilli with E2, P4 or MPA would alter the response of GECs to subsequent HIV-1 exposure. GECs were grown to confluence in the presence or absence of E2, P4, and MPA, then treated with the probiotic RC-14, and exposed to HIV-1. Compared to HIV-1 exposure alone, TERs for GECs exposed to probiotic lactobacilli reached levels equal to control GECs that had no exposure to HIV-1. In contrast, GECs treated with or without hormones and challenged with HIV-1 had significantly reduced TER (Figure 5A). The probiotic lactobacilli treatment had a positive effect on the TER of GECs exposed to HIV-1 regardless of prior exposure to no hormone, E2 or P4 (Figure 5A). As a further assessment of barrier permeability, pre-treatment with RC-14 prior to HIV-1 exposure abrogated the HIV-1-mediated increase in FITC-dextran leakage in no hormone, P4 and MPA treated GECs (Figure 5B). 

Pre-treatment with RC-14 significantly reduced HIV-1-mediated induction in TNF-α and IL-1α in the presence of no hormone or P4 (Figure 5C,D). GECs grown in the presence of E2 and MPA and pre-treated with RC-14 did not show a significant decrease in TNF-α compared to HIV-1 exposure alone (Figure 5C). However, it should be noted that in the presence of E2 or MPA, HIV-1 did not induce a significant TNF-α response in comparison to the no treatment control or P4 (Figure 5C). This indicates that the effect of probiotic lactobacilli on HIV-1-mediated increases in TNF-α is only demonstrated clearly when GECs are pre-treated with P4. In contrast, a significant increase in production of IL-8 was only observed in HIV-1 exposed cells that had previous exposure to MPA (Figure 5E). In GECs grown in MPA, the added treatment with RC-14 resulted in reduced IL-8 response to HIV-1, indicating that probiotic lactobacilli could reverse the effects of MPA alone. There was a significant increase for all hormonal conditions in response to HIV-1 alone. However, pre-treatment with RC-14 lactobacilli prior to HIV-1 exposure significantly reduced the HIV-1-mediated increase in RANTES production for all hormone treatment groups (Figure 5F). 

In order to confirm the changes in GEC monolayer TER and permeability, we performed fluorescent staining for the tight junction protein ZO-1 with visualization by confocal microscopy (Figure 6). Representative images and quantitative results shown in Figure 6 indicate that there is a decrease in ZO-1 observed after HIV-1 treatment of GECs that were pre-treated with no hormone or with P4 or MPA. GECs grown in the presence of E2 and exposed to HIV-1 showed reduced impairment of ZO-1 expression. Furthermore, pre-treatment with RC-14 prior to HIV-1 exposure shows a maintenance of ZO-1 lattice structure compared to hormone with HIV-1 exposure alone. 

## 4. Discussion

The results presented here indicate that irrespective of hormonal conditions, two probiotic lactobacilli strains enhanced GEC barrier function and diminished the proinflammatory mediators produced by GECs in response to HIV-1 exposure. Among the hormones we tested, E2 and MPA demonstrated reduced production of proinflammatory cytokines and chemokines by GECs, an effect that was sustained even after HIV-1 exposure. However, the attenuating effect of E2 and MPA was not evident for production of the proinflammatory chemokine RANTES, suggesting that these cytokines and chemokines may have differential pathways of regulation. The increase in barrier function observed with lactobacilli treatment and the decrease of inflammation observed in the presence of E2 indicate that their combination may create an environment within the FGT that is important for protection against HIV-1.

An inflammatory environment in the FGT is associated with a disruption in epithelial barrier function, which increases permeability between adjacent cells allowing microbial translocation of HIV-1 [11,44], and elevates production of proinflammatory TNF-α, causing further barrier damage [11]. Previous literature has indicated that both lactobacilli and E2 have beneficial effects in the FGT and have been implicated in protection against HIV-1 [8]. However, whether these beneficial effects are mediated through their effects on the mucosal epithelial cells lining the upper genital tract is unclear. Therefore, we undertook this study to examine the direct effects of probiotic lactobacilli, hormones or HC on the response of GECs prior to and post-HIV-1 exposure.

Given that both inflammation and a decrease in barrier function are associated with increased HIV-1 acquisition and disease progression [11,44,45,46], we sought first to explore the role of hormones, hormone-based contraceptives and lactobacilli on the barrier functions of primary cultures of human GECs. Using our GEC culture system, we showed that probiotic strains of lactobacilli (GR-1 and C-14) significantly increased TER, a measure of barrier integrity (Figure 1A) and significantly decreased barrier permeability (Figure 1B). The sex hormone E2 enhanced barrier integrity but did not reduce barrier permeability, whereas treatment with MPA reduced integrity and increase barrier permeability. This ability to enhance barrier functions could explain why a lactobacilli-dominated vaginal ecosystem is protective against pathogens such as HIV-1, and why application of probiotic strains may offer this same advantage.

The presence of either strain of *Lactobacillus* in the absence of HIV-1, did not result in any change in production of TNF-α, IL-1α, IL-8 or RANTES by GECs. On the other hand, sex hormone E2 exerted a significant anti-inflammatory effect on GECs, as seen by a reduction in the secretion of TNF-α, IL-1α, and IL-8 (Figure 2), whereas P4 and MPA had significant effects on reducing IL-1α alone. Neither E2, P4 nor MPA altered RANTES production by GECs. The overall suppressive activity of E2 and its selective ability to reduce TNF-α, a cytokine well known for its ability to induce barrier damage, indicates E2 could have a dominant effect in reducing local inflammation. Given that inflammation is a known contributor to STI acquisition in the FGT, the presence of E2 may play a protective role against some pathogens. 

The finding that probiotic lactobacilli can increase epithelial barrier function agrees with studies of both intestinal and genital tract systems [47,48]. The ability of lactobacilli to reduce pathogen-induced inflammation also aligns with other studies [47,49]. Interestingly, anti-inflammatory effects are also conferred by the lactic acid these strains produce [48]. By lactobacilli application strengthening these host defenses, relatively quickly (within two hours) making it more difficult for HIV-1 to translocate the GECs, it suggests a practical means by which women might be able to reduce their risk of infection. Although lactobacilli had no direct effect on GECs in regard to their normal production of inflammatory cytokines and chemokines (Figure 1), pre-treatment with lactobacilli did greatly reduce that production of TNF-α, IL-1α, and RANTES in response to subsequent HIV-1 challenge (Figure 4). Thus, a relatively brief exposure (2 h) to probiotic lactobacilli primes GECs to increase barrier function and reduce their inflammatory mediator response to HIV-1, indicating a key role for these common FGT bacteria in conferring resistance to HIV infection.

Estrogen has been applied directly to the vagina of post-menopausal women for many years. Studies showed it can reduce the incidence of recurrent urinary tract infection and increase production of antimicrobial lacto-transferrin in the vagina [49,50]. The present study showed that E2 can increase barrier function as well as decrease production of proinflammatory TNF-α and IL-1α (Figure 2). A proinflammatory environment in the FGT, including increased IL-1α and IL-1β, has been associated with BV and increased acquisition of STIs [50]. By countering TNF-α that is released in response to HIV-1 and the damage it causes to the integrity of the epithelial cell barrier, it may be possible for estrogen to interfere with translocation of HIV-1 across the epithelium [11]. This is supported by E2 significantly reducing leakage by GEC cultures and TNF-α and IL-1α production when challenged with HIV-1. In contrast, the apparent dichotomous observation of greatly increased barrier dysfunction and leakage along with decreased TNF-α and IL-1α production after treatment with MPA and HIV-1 challenge indicate that MPA may still lead to enhanced HIV-1 infection due to increased viral entry. 

Regardless of hormone (E2 vs. P4) pre-treatment, the addition of *L. reuteri* RC-14 improved barrier integrity (TER) in GECs challenged with HIV-1 indicating that this barrier enhancement was largely dominated by the effects of lactobacilli (Figure 6). Likewise, HIV-1 mediated induction of TNF-α and IL-1α was greatly reduced by the combination of E2 or MPA with RC-14. 

The exact mechanism by which lactobacillus treatment induces a positive effect on the GEC barrier function, even in the presence of HIV-1, remains to be examined. However, it is well established that HIV-1 binds to TLR-2 and TLR-4, thus activating the MAPK, PI3K, and NF-κB transcriptional pathways, resulting in up-regulation of proinflammatory cytokines [12,51]. Subsequently, the increased proinflammatory cytokines result in further activation of myosin light chain kinase (MLCK) and myosin light chain (MLC) phosphorylation and cytoskeleton contraction, which induce delocalization of ZO-1 from tight junctions, thus causing disassembly junctional proteins [11,52]. This damage to barrier function allows pathogens to cross the epithelium. In vitro studies using intestinal epithelial cells suggests that lactobacilli increase barrier function via tight-junction protein expression mediated by the TLR-2 pathway [53]. The finding that *L. reuteri* RC-14 better maintained ZO-1 lattice structure countering HIV-1’s effect on the protein again supports the protective properties of lactobacilli. 

Although both E2 and MPA appeared to have anti-inflammatory effects in our in vitro experiments, the mechanism underlying this effect and the outcomes of the two hormone treatments may be very different in the context of HIV-1 transmission. E2 has been shown in other cells to exert its anti-inflammatory effects through ERα and downregulation of NFkB-p65 expression [54]. E2 has also been shown in a number of studies to protect against HIV-1 and other STIs [3,13,55]. DMPA on the other hand has been shown to exert overall immunosuppressive effect through the GR receptor. Other studies have shown that that MPA treatment of primary endometrial GECs significantly upregulated genes for CCL20 and CXCL8 chemokines but did not lead to differential expression of genes related to the classical inflammatory cytokines [25]. A dichotomized effect of MPA were seen in our current studies where it enhanced barrier leakage significantly, while still exerting anti-inflammatory effects. Thus, while E2 was correlated with truly beneficial effects overall on GECs, MPA in our study appeared to have a mixed effect and is unlikely to enhance the overall protection against HIV-1. This is supported by previous studies that have shown that women using DMPA are at higher risk of HIV-1 acquisitions compared to women not using any HC [19,21,22,23,24,25]. These studies show that MPA increases chemokine (RANTES) levels resulting in enhanced recruitment of target cells, as well as increasing expression of HIV-1 co-receptors such as CCR5 and CXCR4. Thus, despite its anti-inflammatory effect, the increased barrier leakage combined with presence of more target cells and increased expression of HIV-1 co-receptors likely contribute to increased HIV-1 susceptibility in women using DMPA [18,56].

Overall, our study provides a rationale to test a combination of E2 with probiotic *Lactobacillus* to enhance vaginal epithelial barrier function, decrease inflammation, and potentially lower HIV-1 acquisition, replication and shedding. These results provide clear insight into how factors in the genital microenvironment can affect HIV-1 acquisition and will subsequently assist in the development of prophylactic strategies to reduce HIV-1 transmission.

## Figures and Tables

**Figure 1 cells-08-01120-f001:**
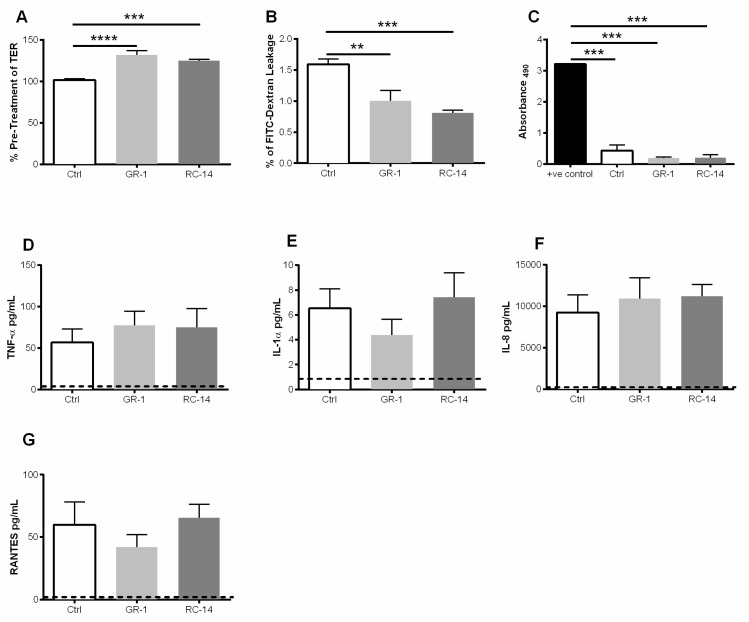
Exposure to probiotic strains of lactobacilli improves barrier function of primary genital epithelial cells and does not alter viability or production of pro-inflammatory cytokines and chemokines. Confluent monolayers of GECs were treated for 2 h with probiotic strains *L. rhamnosus* GR-1 and *L. reuteri* RC-14 abbreviated GR-1 and RC-14 respectively and/or no bacteria (Ctrl). After treatment, non-adherent lactobacilli were removed, and GECs were monitored for a further 24 h. (**A**) TER was measured pre-treatment and at 24 h post-treatment with lactobacilli. TER is expressed as percent of pre-treatment TER. (**B)** Following treatment with bacteria, primary GECs were incubated with FITC-dextran dye on the apical surface, and FITC-dextran leakage in the basolateral compartment was measured after 24 h. Data are expressed as a percentage of FITC-dextran added to the apical compartment. (**C**) LDH release (Abs_490_) by GECs 24 h after treatment with GR-1, RC-14 or no bacteria, or from GEC cultures lysed to release total LDH (positive control). (**D**–**G**) 24 h post-treatment with lactobacilli, cell supernatants were collected and analyzed for the presence of TNF-α, IL-1α, IL-8, and RANTES. Dashed line indicates assay limit of detection. Data shown are representative of 6–9 separate experiments done on tissues taken from 6-9 different donors and shown as mean ± SEM. Each experiment had 3 replicate cultures for each experimental condition. Statistical significance is indicated as: **** *p* < 0.0001, *** *p* < 0.001, ** *p* < 0.01.

**Figure 2 cells-08-01120-f002:**
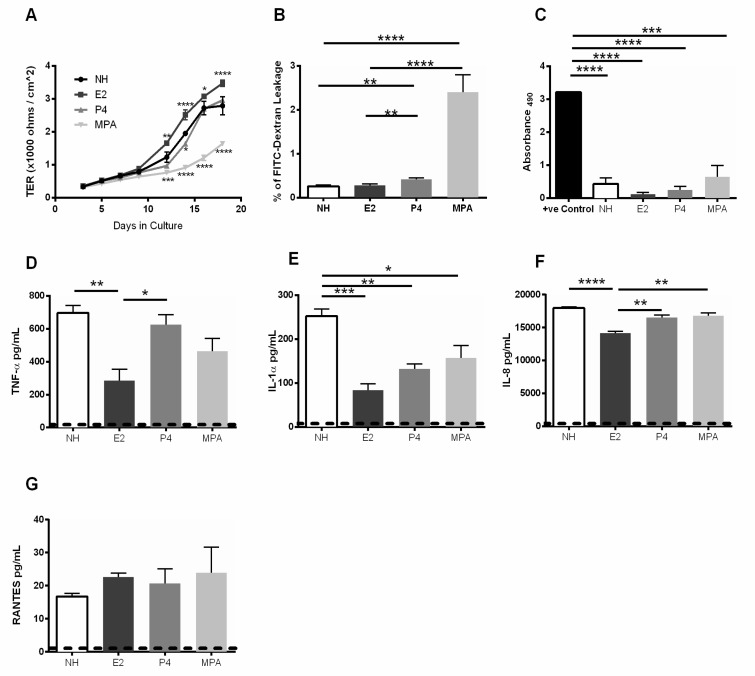
Estrogen treatment reduces barrier leakage and decreases pro-inflammatory cytokine response in primary genital epithelial cells. Primary GECs were grown in the absence (NH = no hormone) or presence of sex hormones E2, P4 or MPA until confluent. (**A**) TER was measured every alternate day starting day 3 post-seeding of cells. (**B**) Following hormone treatment in confluent GEC monolayers, cells were incubated for 24 h with FITC-dextran dye applied to the apical side. 24 h later, 50 µL of basolateral supernatant was collected and FITC-dextran dye was measured using a microplate reader. Dextran leakage was expressed as a percentage of FITC-dextran added to the apical compartment. (**C**) Supernatants taken from confluent monolayers grown in presence of hormone and contraceptive were assessed for LDH release to determine cell stress and cytotoxicity. (**D**–**G**) Supernatants from GECs grown in the presence of E2, P4 or MPA were collected and analyzed for the presence of TNF-α, IL-1α, IL-8, and RANTES. Dashed line indicates lowest detection limit of individual cytokine and chemokine in an assay. Data shown is representative of 6–9 separate experiments done on 6–9 different tissues taken from different donors and shown as mean ± SEM. Each experiment had 3 replicate cultures for each experimental condition. Statistical significance is indicated: **** *p* < 0.0001, *** *p* < 0.001, ** *p* < 0.01, * *p* < 0.05.

**Figure 3 cells-08-01120-f003:**
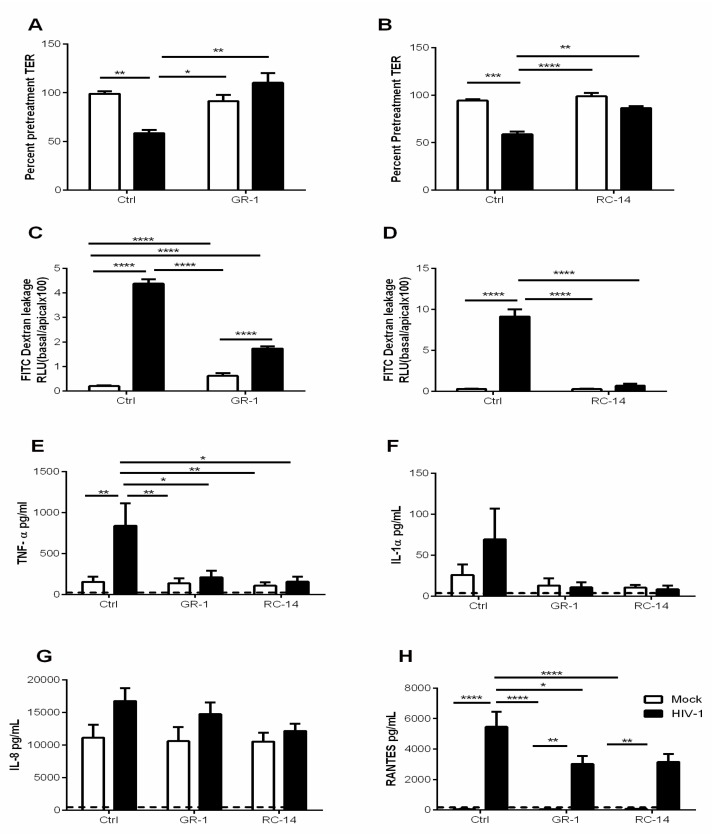
Probiotic strains of lactobacilli ameliorate HIV-1-mediated barrier disruption and downregulate proinflammatory cytokine production by GECs. GECs were treated for 2 h with probiotic strains *L. rhamnosus* GR-1 and *L. reuteri* RC-14 abbreviated GR-1 and RC-14 respectively or no bacteria (Control). Following treatment, various cultures of GECs were then exposed to HIV-1 or no virus (Mock) and monitored for 24 h. (**A**,**B**) TER values were observed pre- and 24 h post-HIV-1 exposure. TER is expressed as percent of TER measured before HIV-1 exposure. (**C**,**D**) Following treatment with lactobacilli and during viral exposure, GEC cultures were treated apically with FITC-labeled dextran dye. At 24 h, 50 µL of basolateral were collected and fluorescence was measured using a microplate reader. The dextran leakage in the basolateral compartment is expressed as a percentage of dextran added to the apical compartment. (**E**–**H**) 24 h post-exposure to HIV-1, cell supernatants were collected and analyzed for TNF-α, IL-1α, IL-8, and RANTES. Dashed line indicates assay limit of detection. Data shown is representative of 6–9 separate experiments done on cells isolated from 6–9 different tissues, each experiment had 3 replicate cultures for each experimental condition. Data is plotted as mean + SEM. Statistical significance is indicated: **** *p* < 0.0001, *** *p* < 0.001, ** *p* < 0.01, * *p* < 0.05.

**Figure 4 cells-08-01120-f004:**
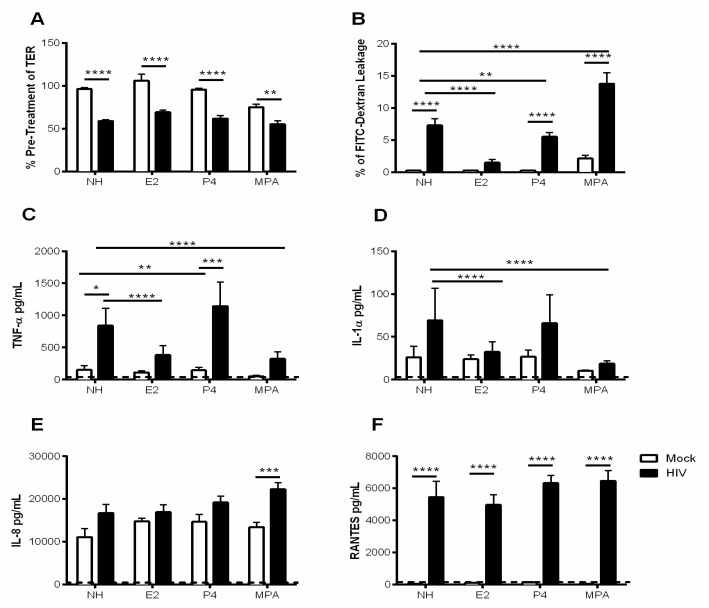
Estrogen treatment prevents HIV-1-mediated barrier disruption in GECs. GECs were grown in the absence (NH) or presence of sex hormones E2, P4 or MPA until confluence was reached. GECs were then exposed to HIV-1 and monitored for 24 h. (**A**) TER values were observed at 0 h and 24 h after HIV-1 exposure and expressed as percent of pre-treatment TER. Data is derived from 3 separate GEC culture experiments, with a minimum of two replicates per experimental condition were included. (**B**) FITC-dextran leakage to basolateral compartment after 24 h exposure to HIV-1. The FITC-dextran leakage is expressed as a percentage of FITC-dextran added initially. (**C**–**F**) 24 h post HIV-1 exposure, cell supernatants were collected and analyzed for TNF-α, IL-1α, IL-8 and RANTES. Dashed line indicates assay limit of detection. Representative data is shown from 5–8 separate experiments done on cells isolated from 5–8 different tissues obtained from different donors; each experiment had 3 replicate cultures for each experimental condition. Data is plotted as mean + SEM. Statistical significance is indicated: **** *p* < 0.0001, *** *p* < 0.001, ** *p* < 0.01, * *p* < 0.05.

**Figure 5 cells-08-01120-f005:**
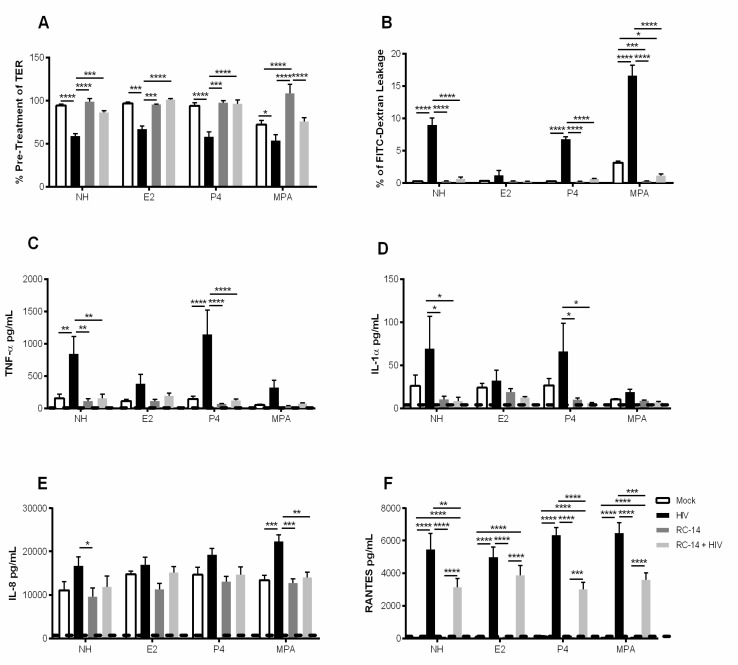
Probiotic lactobacilli and estrogen ameliorate HIV-1-mediated barrier disruption. GECs were grown in the absence of hormone (NH) or with E2, P4 or MPA until confluence was reached. Primary GEC cultures were treated for 2 h with *L. reuteri* RC-14 or medium control (Mock). After lactobacilli treatment GECs were exposed to HIV-1 and monitored for 24 h. (**A**) TER values were observed at 0 and 24 h post-viral exposure and expressed as percent of pre-treatment TER. (**B**) During viral exposure, primary GECs were incubated with an apical treatment of FITC-dextran dye for 24 h. The FITC-dextran leakage into the basolateral compartment is expressed as a percentage of total FITC-dextran added to the apical compartment. (**C**–**F**) 24 h following HIV-1 exposure, cell supernatants were collected and analyzed for TNF-α, IL-1α, IL-8, and RANTES production. Dashed line indicates assay limit of detection. Data are representative of 5–8 experiments from 5–8 different tissues, each tissue obtained from a separate donor. Each experiment had 3 replicate cultures for each experimental condition. Data is plotted as mean + SEM. Statistical significance is indicated: **** *p* < 0.0001, *** *p* < 0.001, ** *p* < 0.01, * *p* < 0.05.

**Figure 6 cells-08-01120-f006:**
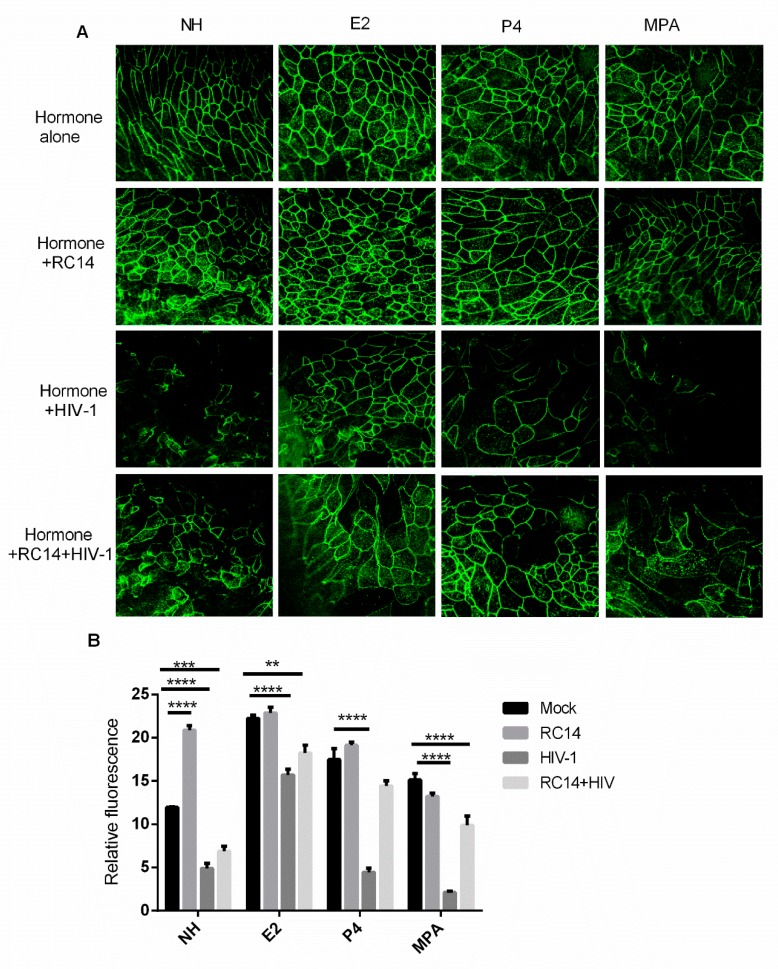
Probiotic lactobacilli and estrogen protect GECs from HIV-1-mediated tight junction degradation. GECs were grown in the presence or absence (NH) of E2, P4 or MPA until confluence was reached. Confluent GECs were pre-treated for 2 h with *L. reuteri* (RC-14) at 100 CFU/cell or no bacteria. After pre-treatment, non-adherent lactobacilli were removed, and primary GEC were exposed to HIV-1. After 24 h, cells were fixed and stained for ZO-1 tight junction proteins. Images were captured with Olympus confocal microscope. Representative images (×1260 magnification) of each culture condition are shown (**A**). Fluorescence of 3 different images taken for each culture condition (see Methods for details) and were measured by Image J software (**B**). Data is plotted as mean + SEM of relative fluorescence. Statistical significance is indicated: **** *p* < 0.0001, *** *p* < 0.001, ** *p* < 0.01.

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
