# Peer review of "Protective Effect of Probiotic Bacteria and Estrogen in Preventing HIV-1-Mediated Impairment of Epithelial Barrier Integrity in Female Genital Tract"

_cells, 2019, doi:10.3390/cells8101120_

Round 1

Reviewer 1 Report

This is a very clear and well-written paper, and the subject matter is very interesting.  I have only a few minor comments/suggestions.

1) In Figures 2, 4, 5, and 6, the authors should list the definition for 'NH" (I assume it means no-hormone control, but I didn't see this in any of the figure legends).

2) Line 120: Can the authors comment on whether the E2, P4, or MPA preparations contained DMSO or other organic solvent, at any concentration?  I assume that the final concentration of DMSO would be very low if, for instance, it were used to make concentrated master stocks of the hormones. If DMSO (or similar) was present, it would suffice to list the final concentration in the assay wells.

3) Line 206: The sentence discussing differences in LDHbetween control and Lacto-exposed cells should end with "(data not shown)".

4) Lines 225, 452: Sentences appear to be missing articles/words (i.e.- of, the).

5) Line 471: Please provide references to support the statement that "women using DMPA are at higher risk of HIV-1 acquisition compared to women not using any HC".  (I assume that the appropriate citations are references 18, 20, 21, 22-24, but they're missing here).

Author Response

REVIEWER #1:

This is a very clear and well-written paper, and the subject matter is very interesting.  I have only a few minor comments/suggestions.

Reply: We thank the reviewer for the positive comments and for appreciating the manuscript.

1.  In Figures 2, 4, 5, and 6, the authors should list the definition for 'NH" (I assume it means no-hormone control, but I didn't see this in any of the figure legends).

Reply:

The definition of NH has been added to the legends of Figures 2, 4, 5, and new figure 6B.

2.  Line 120: Can the authors comment on whether the E2, P4, or MPA preparations contained DMSO or other organic solvent, at any concentration?  I assume that the final concentration of DMSO would be very low if, for instance, it were used to make concentrated master stocks of the hormones. If DMSO (or similar) was present, it would suffice to list the final concentration in the assay wells.

Reply:

There was no use of DMSO in any stock hormone preparation.  Cell culture compatible water-soluble estradiol and progesterone were used in our experiments. A more detailed description of each hormone stock and dilution is now included in the methods section 2.2.  Ethanol was used as a solvent only for MPA and its final dilution has now been included in methods.

3.  Line 206: The sentence discussing differences in LDH between control and Lacto-exposed cells should end with "(data not shown)".

Reply:

In fact, the data are shown in Figure 1C. We have added “(Figure 1C)” to the text to clarify (Line 231, pg. 6)

4.  Lines 225, 452: Sentences appear to be missing articles/words (i.e.- of, the).

Reply:

We thank the reviewer for finding these editing errors. Minor corrections to grammar have been done in these locations and a few others.

5.  Line 471: Please provide references to support the statement that "women using DMPA are at higher risk of HIV-1 acquisition compared to women not using any HC".  (I assume that the appropriate citations are references 18, 20, 21, 22-24, but they're missing here).

Reply:

The appropriate list of references is now included at the end of that sentence in the discussion (Line 522).

Reviewer 2 Report

In this manuscript Dizzell et al. use their GEC model to test if probiotic bacterial strains can boost epithelial integrity. Not surprisingly, they find that probiotic bacterial strains boost epithelial integrity measured in terms of TEER and FITC-dextran leakage. Without going in to any mechnistic insights or using in vivo models, this study is only an in vitro description of what is mostly known and expected.

Specific points

Although it is far from a consensus, authors should cite the primary papers where it was shown that transition zone between ecto and endo cervix is highly susceptible to HIV-1 infection, not only the reviews written by themselves: “Studies have demonstrated that upper genital tract, especially the transition zone between ecto and endo cervix is highly susceptible to HIV-1 infection [3,8].” Lactate dehydrogenase (LDH) is produced by dead or dying cells and measured in culture supernatants as an indicator of cellular cytotoxicity and cytolysis”.

Actually, LDH is found/produced by most humans cells, all the time, it is only released into the supernatant when cells lose membrane integrity. Please correct.

Also authors should acknowledge the seminal work done by D. Kwon’s group and others in this field. Are authors are implying that a 1.5 times change in biologically insignificant for one cytokine (TNF, Fig 1D) while highly significant for another cytokine (IL8, Fig 2F? How do they explain difference in the range of 10x in cytokine concentrations in mock/control groups of GECs (for example, TNF, Fig 1D vs Fig 2D). If data is obtained from n=6-9 samples, as claimed, what’s the explanation for the huge variation?

Do these concentrations correspond with what is observed in healthy/diseased conditions, in lavage or blood etc.

More information should be provided about imaging. Just mentioning, “Images were captured with Olympus confocal microscope” is not sufficient what was the objective, which dyes, laser, excitation, emission, filter etc. were used?

This journal does not impose a strict word limit, so they should include detailed methods for each experiment.

Fig 8. Are these single plane confocal sections or maximum intensity projections? What’s the take home message of these images? Why don’t they show any quantification from these data An important point that is missing from the study is that does the probiotic bacteria directly or a secreted bacterial product affect HIV? Given the shortcomings of in vitro GEC model, the study would have acquired more relevance had the authors tested the effect of these bacteria on HIV transmission in their humanized mouse models (Nguyen et al.)

Author Response

REVIEWER #2

In this manuscript Dizzell et al. use their GEC model to test if probiotic bacterial strains can boost epithelial integrity. Not surprisingly, they find that probiotic bacterial strains boost epithelial integrity measured in terms of TEER and FITC-dextran leakage. Without going in to any mechanistic insights or using in vivo models, this study is only an in vitro description of what is mostly known and expected.

Specific points

1)    Although it is far from a consensus, authors should cite the primary papers where it was shown that transition zone between ecto and endo cervix is highly susceptible to HIV-1 infection, not only the reviews written by themselves: “Studies have demonstrated that upper genital tract, especially the transition zone between ecto and endo cervix is highly susceptible to HIV-1 infection [3,8].” 

Reply:

The reference for the original study (Pudney et al.) defining the ecto-endo cervix transition zone and demonstrating presence of significant population of CD4 target cells has been added (line 50, reference 9)

2)    Lactate dehydrogenase (LDH) is produced by dead or dying cells and measured in culture supernatants as an indicator of cellular cytotoxicity and cytolysis”. Actually, LDH is found/produced by most humans cells, all the time, it is only released into the supernatant when cells lose membrane integrity. Please correct.

Reply:

The reviewer provides a valid point regarding LDH production by normal cells.  We have corrected the statement (pg. 6, line 226) to read: “…(LDH) is released into the supernatant when cell membrane integrity is compromised and ….” Thank you for pointing out this oversight.

3)    Also, authors should acknowledge the seminal work done by D. Kwon’s group and others in this field. Are authors are implying that a 1.5 times change in biologically insignificant for one cytokine (TNF, Fig 1D) while highly significant for another cytokine (IL8, Fig 2F)?

Reply:

The appropriate reference to the work of Dr. Kwon’s group has now been included in the introduction paragraph discussing the effects of lactobacilli. (pg. 2, line 85, reference 32).

Regarding the significant differences between groups as presented in Figures 1 and 2, any reference to significance in the figures has been made in the context of statistical differences between groups.  So, TNF responses in Figure 1 were not statistically significant among groups, but IL-8 responses were significantly different between groups in Figure 2. We are not suggesting that statistically significant differences directly imply biological differences in function. Biological significance would require functional measures of the cytokines produced using in vivo studies that are beyond the scope of this work.  Therefore, our results are indicators of how GECs respond in vitro and can only suggest but not prove any biological significance.

4)    How do they explain difference in the range of 10x in cytokine concentrations in mock/control groups of GECs (for example, TNF, Fig 1D vs Fig 2D). If data is obtained from n=6-9 samples, as claimed, what’s the explanation for the huge variation?

Reply:

The reviewer has raised a valid point. The data shown in most figures is actually representative data from 6-9 different experiments done on 6-9 separate tissues obtained from individual donors. Within each experiments, each experimental condition was run in at least 3 replicate cultures. We apologize for the confusion from the way this was stated in the figure legends. This has now been corrected throughout the figure legends. The reason that we frequently cannot pool data from different tissues is because the baseline cytokine secretion profile can be very different among different donors, even though the response of each tissue to specific experimental treatment is similar. The reason for these differences are not known, they could be inherent biological differences among different women, but could also be due to medical conditions for which the hysterectomy is being performed. As best as can be determined from the pathology, the tissues piece we obtain from which epithelial cells are isolated are normal in appearance, but any biological changes in functions cannot be determined by gross examination. Given the significant variability in cytokine production, we cannot pool data from all the different tissues. These baseline differences are exactly what the reviewer has noted and is pointing out in the different figures (Fig 1D vs Fig 2D) where baseline secretion of TNF in one tissue can be 10 times lower than another tissue. However, regardless of baseline levels of TNF, treatment with HIV increased TNF levels in cultures grown from tissues from different donors while treatment with E2 decreased TNF secretion. The data shown in each figure is a representative from a number of experiments where the results were similar.  We have published a number of papers using this methodology and our results have been reproduced by a number of other labs (Nazli et al, PLOS Pathogens, 2010, Nazli et al, J. Immunol. 2013, Ferreira et al, JID, 2014, Nazli et al, Cell Mol Immunol 2017). We have changed the figure legends to state more explicitly these details and also added a statement regarding baseline differences in cytokines to results section 3.1 (line 240). We hope this clarifies this issue for the reviewer.

5)    Do these concentrations correspond with what is observed in healthy/diseased conditions, in lavage or blood etc.

Reply:

There is no reason to expect the results from in vitro pure epithelial cell preparations from tissues will have a similar range in concentration to that of human lavage or plasma samples. The CVL would more likely reflect local tissue production of cytokines from all different cells, to which epithelial cells could contribute significantly, but there is no reason to suggest that must be so. We looked at published data (extracted table from supplementary data, reference below) regarding CVL and plasma cytokines from normal uninfected individuals. We see that, not surprisingly, the cytokine concentrations in CVL are not in the same concentration range as our in vitro cultured GECs, as expected.  We note however that the ratio of TNF-a to IL-8 in normal CVL is similar to that of our in vitro GEC experiments, where IL-8 is always much higher than TNF-a, showing some similarity, and perhaps indicating that genital epithelial cells do contribute significantly to the cytokines found in CVL. However, it is impossible to directly correlate levels of cytokines secreted in vivo by complex genital tract tissue with a few hundred thousand pure epithelial cells grown in culture.

CVL expression

Plasma expression

jhdbkahdkhaudhau Cytokine (pg/ml)

HIV-1 uninfected CSW

P value

N=49

Il-1a

278.0 (729.7)

1.8 (5.1)

<0.0001

Il-1b

45.9 (127.3)

0.1 (0.4)

<0.0001

TNF-a

1.9 (4.2)

7.0 (4.0)

<0.0001

IL-6

16.2 (33.5)

1.2 (4.5)

<0.0001

IL-8

1402.0 (2131.0)

39.4 (62.7)

<0.0001

From: Lajoie J, Juno J, Burgener A, Rahman S, Mogk K, Wachihi C, Mwanjewe J, Plummer FA, Kimani J, Ball TB, Fowke KR. A distinct cytokine and chemokine profile at the genital mucosa is associated with HIV-1 protection among HIV-exposed seronegative commercial sex workers. Mucosal Immunol. 2012 May;5(3):277-87. doi: 10.1038/mi.2012.7. Epub 2012 Feb 8.

6)    More information should be provided about imaging. Just mentioning, “Images were captured with Olympus confocal microscope” is not sufficient what was the objective, which dyes, laser, excitation, emission, filter etc. were used? This journal does not impose a strict word limit, so they should include detailed methods for each experiment.

Reply:

We have added the imaging method details as requested to the Methods section 2.8 (pg. 4, Line 168-176) and figure 6 legend (line 425-428)

7)    Fig 8. Are these single plane confocal sections or maximum intensity projections? What’s the take home message of these images? Why don’t they show any quantification?

Reply:

Yes, these are single plane confocal sections. As suggested by the reviewer, quantification of fluorescence was performed on images with Image J software and a graph of this data has been added as a new Figure 6B (pg. 13) and commented on in Results section 3.5 (line 413)

8)    An important point that is missing from the study is that does the probiotic bacteria directly or a secreted bacterial product affect HIV?

Reply:

This is an interesting question, and worthy of further studies in the future, where the mechanism of bacterial interactions with GEC could be examined.  It is known that the lactobacilli attach to epithelial cells.  However, while clinical studies show probiotics can modulate inflammation in the FGT (some references below) in HIV infected patients, none have distinguished bacterial interaction or bacterial products as necessary for the effects on FGT inflammation. In studies not related to HIV, short chain fatty acids such as butyrate secreted by the commensal bacteria can play an anti-inflammatory role and enhance the barrier function of epithelial cells.             

d’Ettorre G. et al. PLoS One, 2015; 10:e0137200 Gori, A. et al. Mucosal Immunol, 2011; 4:554-563 Kim C.J., et al. HIV clinical trials; 2016; 17:147-157 Liu, Y., et al. J Clin Pharm; 2018; 10.1002/jcph.1121 5. Bach Knudsen,  K,E, Nutrients. 2018 10(10): 1499.

9)    Given the shortcomings of in vitro GEC model, the study would have acquired more relevance had the authors tested the effect of these bacteria on HIV transmission in their humanized mouse models (Nguyen et al.)

Reply:

We agree with the reviewer that the in vivo model would be an important and necessary study to test the functional significance of our in vitro observations.  However, that work would be extensive and beyond the scope of this paper that sought to provide evidence of the combined effects of lactobacilli and hormones that would be relevant to HIV infection at the level of the epithelial barrier function. We are very interested in exploring this line of investigation in our future studies.

Round 2

Reviewer 2 Report

I don't have any comments